# Genotoxic Risks to Male Reproductive Health from Radiofrequency Radiation

**DOI:** 10.3390/cells12040594

**Published:** 2023-02-12

**Authors:** Puneet Kaur, Umesh Rai, Rajeev Singh

**Affiliations:** 1Department of Environmental Studies, Satyawati College, University of Delhi, Delhi 110052, India; 2Department of Zoology, University of Delhi, Delhi 110007, India; 3Department of Environmental Science, Jamia Millia Islamia (A Central University), New Delhi 110025, India

**Keywords:** radiofrequency radiation, genotoxicity, DNA damage, male infertility, oxidative stress, reproductive health

## Abstract

During modern era, mobile phones, televisions, microwaves, radio, and wireless devices, etc., have become an integral part of our daily lifestyle. All these technologies employ radiofrequency (RF) waves and everyone is exposed to them, since they are widespread in the environment. The increasing risk of male infertility is a growing concern to the human population. Excessive and long-term exposure to non-ionizing radiation may cause genetic health effects on the male reproductive system which could be a primitive factor to induce cancer risk. With respect to the concerned aspect, many possible RFR induced genotoxic studies have been reported; however, reports are very contradictory and showed the possible effect on humans and animals. Thus, the present review is focusing on the genomic impact of the radiofrequency electromagnetic field (RF-EMF) underlying the male infertility issue. In this review, both in vitro and in vivo studies have been incorporated explaining the role of RFR on the male reproductive system. It includes RFR induced-DNA damage, micronuclei formation, chromosomal aberrations, SCE generation, etc. In addition, attention has also been paid to the ROS generation after radiofrequency radiation exposure showing a rise in oxidative stress, base adduct formation, sperm head DNA damage, or cross-linking problems between DNA & protein.

## 1. Introduction

Radiation is a series of energy that flows through the medium in the form of atomic or subatomic particles and, as electric and magnetic waves form. Depending on the particle’s energy, radiation is characterized as ionizing (IR) and non-ionizing (NIR). Ionizing radiation is considered to be more deleterious than non-ionizing radiation due to its high emission properties to break the bonds and knock out the electrons from its molecular shell. This type of radiation produces severe harm to the biological system leading to DNA impairment and tissue damage. X-rays (3 × 10^16^ Hz), gamma rays (>10^19^ Hz), and alpha particles (8–12 Hz) are the forms of ionizing radiation with sufficient frequency to cause such disruption to the living system [1]. On the other hand, extra-low frequency (ELF) (0.1 Hz–1 KHz) and radio frequency (RF) (10 MHz–300 GHz) are the forms of non-ionizing radiation that do not have enough quantum energy to break the molecular bonds [2]. Despite deliberating less intensity, non-ionizing radiation was found to create substantial health problems including cancer risks that need to be discussed further at the genomic level [3,4,5].

RFR-EMF is considered as one of the possible carcinogen sources to humans under group ‘2B’ category according to the International Agency for Research on Cancer [6]. Radiofrequency radiation is produced by man-made wireless radiowaves or microwaves products such as transmission lines (50–60 Hz), microwave ovens (2.45 GHz), laptops and Wi-Fi (2.4 GHz), computer monitors (60–90 Hz), AM radio transmissions (530–1600 KHz), FM transmissions (50–70 MHz) and mobile phones (850 MHz–2.4 GHz) [2]. Radiations emitted from RFR devices display antagonistic effects on the biological system covering the central nervous system (sleep disturbances, headache disorder), circulatory system (increased heart rate and blood pressure), and reproductive system (male and female fertility issues) [2,6,7,8]. It can affect the living body by two mechanisms: thermal and non-thermal. Thermal effects cause tissue heating by increasing the body temperature by more than 1 °C. Due to the body’s inability to emit away the excessive heat absorbed, the thermal mechanism poses cell function impairment due to conformational changes in the heat shock proteins (hsp) or stress response proteins [9]. In contrast, non-thermal effects underwent disruption of cell membrane integrity by raising the body temperature below 1 °C [9,10]. These cellular changes have been reported to result in endothelial dysfunction, alterations in the blood–brain barrier, compromised immune system, changes in the cell signaling pathway, and nervous system disorder [11,12,13,14,15]. However, studies have also been enlightened with minor biological problems created through non-thermal RFR exposure [16,17,18].

To measure the effects of radiofrequency radiation, a standardized unit called specific absorption rate (SAR) is used to find the rate of energy absorbed per unit mass in the body, expressed as watt/kg. The safe dose for whole-body exposure is recommended to be 2.0 W/kg. However, according to WHO (world health organization), the given SAR lethal dose limits up to 4.0 W/kg [19,20]. Polarization, frequency, conductivity, density, exposure time, and distance are essential extrinsic or intrinsic parameters depending on the proportion of SAR absorbed by a living tissue [21].

The male reproductive system is one of the most affected biological systems, reported due to organ (testes) sensitivity to RF-EMF (Figure 1) [22,23,24]. Morphological changes in testicular tissue, decreased sperm count, increased mortality, disrupted sperm DNA integrity, or increased permeability of the blood–brain barrier along with increased mitochondrial ROS production considered as the unexpected events reported so far due to the power density and frequency of cell phones, which might be responsible for male infertility under the influence of oxidative stress [25,26,27,28,29,30,31].

Long-term RFR exposure generates excessive reactive oxygen species, which may alter the endocrine mechanism of the male reproductive system. In this context, leydig cells are prime prudential interstitial cells under constant exposure to RFR [32]. Leydig cells are responsible for producing 95% of testosterone by supporting spermatogenesis in the male body under the stimulation of luteinizing hormone (LH) [33]. Persistent mobile phone exposure was reported to decrease the serum testosterone levels affecting sexual differentiation of the fetus as well as male spermatogenesis [27,34]. A study has also been evident for the upregulation of the Est1 oncogene in mouse leydig cells disrupting leydig cell function [35]. Despite having these published data, some articles suggest insignificant cellular toxicity even after acute or chronic RFR exposure conducted in vitro or in vivo [36].

Besides cell toxicity, genotoxicity is the primary area of concern, considered as one of the key biological indicators of carcinogenicity risk under the influence of RFR exposure (Figure 1) [37].

Systemic reviews by WHO have also been conducted among several evidence streams explaining the adverse health risks associated with RF-EMF exposure, except for the experimental studies concerning genomic effects at cellular (in vitro) level [38]. Romeo et al. [39] has explained further with the systemic review of studies presenting the potential of inducing genetic effects by RF-EMF in a mammalian in vitro cell model. However, RFR genotoxicity in the male reproductive system has still remained elusive despite much research and varied assessment. Therefore, to evaluate further, this review aims to assess the genotoxic effects of radiofrequency radiation on the male reproductive system, underlying male infertility issues in both in vitro and in vivo models, along with a focus on oxidative stress after exposure.

## 2. Literature Search and Methodology

The data was collected and analyzed via computerized database search such as PubMed, Google Scholar, and Science Direct to review the genotoxic effects of radiofrequency radiation exposure on male reproduction. The literature search was conducted by entering keywords such as ‘Leydig cell and radiation’, ‘Genotoxicity and RFR exposure’, ‘Genotoxic impact on male fertility and non-ionizing radiation’, ‘RFR induce male fertility’, or Radiofrequency radiation and DNA damage. All the articles published till May 2022 were incorporated in the study. Additional literature articles were collected from the Web of Science site to explore further.

All the published data, research articles, and guidelines were included in the presenting document, covering both in vivo and in vitro studies assessing genotoxicity.

## 3. RFR-Induced Genotoxicity on Male Reproduction

DNA integrity is the utmost concern for a cell concerning infertility. Usage of mobile phones as a radiofrequency (RF) exposure source in close vicinity to the gonads escalates possible repercussions on the male reproductive system [40]. Genotoxic studies deal with the changes that occur in the DNA of the cells at the molecular level during the controlled biological events of the organism. Many conventional methods are used to assess these studies, which include comet assay, micronucleus assay (MN), chromosomal assay (CA), or the detection of sister chromatid exchange (SCE). The potential effects of EMR on the genetic material of the cells are dominant enough to create genotoxic effects, confining damage to germ cells with respect to mutations in the next or subsequent generations.

Genetic studies displaying the adverse effects of RFR are conducted with the help of in vitro and in vivo experiments. Many in vitro and in vivo studies on genotoxic effects have been summarized so far, concluding the genomic instability with an increase in DNA fragmentation, chromosomal aberrations, and induction of micronuclei after to RFR [41,42,43,44,45,46,47,48,49,50,51]. At the same time, controversial articles have also been reported, suggesting insignificant DNA effects with in vitro studies [52,53,54,55,56].

Leydig cells have been reported to be the most susceptible cells to EMR, and damage to these cells may affect spermatogenesis [57]. Due to possible alteration in testosterone receptors, PKC enzyme complex, oxidative stress, mRNA expression for P450 cholesterol side-chain lyase (the first enzyme in steroidogenesis), and maturation arrest in the spermatogenesis; in vivo findings reported induction of DNA damage to spermatozoa and leydig cell [58,59,60]. Additionally, cells respond to the burden of DNA damage by apoptosis and necrosis. A study by Kesari and Behari [61] reported increased apoptosis in leydig cells after exposure to microwave at 2.45 GHz and 0.11 W/Kg of SAR on 35 days of exposure. At the same time, Aitken et al. [25] concluded with no apoptotic activity in response to induced genetic damage. Studies with the facts investigated that EMF energy is not sufficient enough to damage DNA directly, but the genotoxic effects could be mediated through an indirect mechanism such as free radical hypothesis or ROS generation [62,63,64,65].

### 3.1. In Vitro Studies

In vitro investigations are the fundamental studies to provide unique information and insight on individual radiation exposure to cells without mimicking the in situ condition of cell–cell interaction within a tissue or between the tissues [66]. Such studies can only contribute to providing data that is potentially obtained without animal and human whole-body exposure and may control confounding variables.

Various data has been reported with an increase in DNA fragmentation after radiofrequency exposure. In this review, we explore the in vitro genotoxic effect of radiofrequency radiation using the following endpoints:

#### 3.1.1. DNA Damage

DNA is the store house for all the genetic content that maintains the vicinity of the cell. As mentioned earlier, Leydig cells and spermatozoa are considered most vulnerable to initiate DNA damage after RF radiation exposure due to loss of antioxidant enzyme capacity and DNA repair function followed by loss of cellular cytoplasm [25]. The induction of such DNA damage may result in poor semen quality and poor fertilization rate- leading to male infertility (Table 1) [67,68].

The majority of men carrying DNA damage and sperm-mortality disturbances are associated with infertility issues [67,69,70]. Several studies have reported sperm DNA damage upon the usage of cell phones in their trouser pockets. Sperm has a limited ability to repair single or double-strand breaks. Additionally, studies with the help of TUNEL assay showed an increment in the sperm DNA integrity defects under the influence of cell phones [30]. In contrast, Falzone et al. [71] did not find any significant DNA damaging effects in the purified sperm sample after EWM exposure using TUNAL assay.

Experiments with mice spermatozoa explained mitochondrial respiratory chain (complex III) as the primitive factor of EMR to cause DNA damage due to oxidative stress [72]. Cultured mouse spermatozoa derived GC-2-cell after receiving RFR at a frequency of 1800 MHz (SAR, 0.13 W/Kg), 1 min per 20 min for 24 h resulted in DNA damage at such exposure intensity [73]. Another study with GC-2-cell has also reported DNA damage at a similar frequency for 24 h under the influence of oxidative stress [74]. However, Duan et al. [75] demonstrated no DNA strand breaks after exposure of mouse spermatocyte-derived G2-2 cells at 1800 MHz for 24 h at GSM talk mode, explained due to insufficient energy to induce such damage in male germ cells directly. Although, the study seemed to be altered after using formamidopyrimidine DNA glycosylase (FPG), which enhanced DNA oxidative damage after RFR exposure at a SAR value of 4 W/Kg. Additionally, treatment with radiofrequency exposure at 1950 MHz presented damaging changes with no oxidative or apoptotic damage [76], while exposure at 850 MHz frequency presented oxidative damage with insignificant DNA damage [77]. Although, investigations reported that under certain conditions like high frequency or high-power intensity; and few cell types (human trophoblast HTR-8/S Vneo cells, human leukocytes, spermatozoa), could display genotoxic effects followed by radiation (RFR) exposure [30,78,79]. However, other controversial studies have conformed to no DNA strand breaks in mouse fibroblast cells, Molt-4 cells, human blood lymphocytes, human ES1 diploid fibroblasts, or Chinese hamster V79 cells under the same exposure conditions [80,81,82,83,84].

Some human studies have also indicated DNA fragmentation in the male germline. De luliis et al. [30] has reported with significant DNA damage (DNA base adducts formation) in human spermatozoa after RFR-exposure to 1.8 GHz frequency, explaining the DNA integrity defects proportional to the exposed SAR (Figure 2). Keeping mobile phones in trouser pockets for a long term has been reported with increased sperm DNA fragmentation after prolonged mobile phone exposure for 3–5 h [85,86,87,88,89,90]. Due to an exponentially increased usage of cell phones, author showed an increase in sperm mortality rate, the activity of sperm acrosin, sperm DNA damage, and seminal clusteine gene expression (CLU), even after 1 h exposure to radiofrequency of 850 MHz with SAR value of 1.46 W/Kg, as compared to the non-exposed control group [91]. Additionally, usage of laptops has been reported to be a causative factor of DNA damage with a progressive decrease in sperm motility [92]. Even combined effects of both smartphone (1800 MHz, 4G) and Wi-Fi (2450 MHz) network reported with human sperm DNA damage with an increase in the percentage of tail DNA and tail moment and decrease in head DNA % in the comet assay along with oxidative damage leading to cause male infertility risk [93]. Such studies implicated potential health effects on male fertility and the wellbeing of their offspring (Figure 2).

#### 3.1.2. Micronuclei and Genomic Instability

Micronuclei (MN) are the small extra-nuclear bodies formed by damaged chromosome acentric fragments in response to clastogenic mutation. Micronuclei are considered a conventional or sensitive biomarker to identify genotoxic effects leading to cell death, chromosomal aberrations, genomic instability, or cancer formation (Figure 2) [94,95].

Radiofrequency radiation may have the ability to induce genotoxic instability and to produce a clastogenic impact on chromatin integrity [13,96]. Additionally, RFR is responsible for inducing aneuploidy in a linear & SAR-dependent manner. Supporting the previous statement, Mashevich et al. [47] reported an increase in aneuploidy via a non-thermal pathway at a frequency of 830 MHz in RFR-exposed cells, as compared to sham exposed. Investigation on cultured rodent cells (V79) also showed positive results after microwave exposure at a frequency of 7.7 GHz with a power density of 0.5 mW/cm^2^, which reported significant destruction in chromosome and micronuclei formation. With an increase in exposure time points (15, 30, 60 min), micronuclei generation showed significant increase (0.043 ± 0.042, 0.050 ± 0.049, 0.073 ± 0.073) in numbers in relation to non-exposed sample (0.016 ± 0.016) [97]. Another study with Chinese hamster lung fibroblast cell line (V79) also supported to induce genotoxicity after 20 h of radiofrequency (RF) exposure at 1950 MHz, SAR (0.15–1.25) W/kg, demonstrating a significant increase in the micronuclei (MN) frequency in the exposed group as compared to the sham control [98]. However, Bisht et al. [99] on the other hand, failed to demonstrate any RF-induced micronuclei formation at a frequency of 835.62 MHz, reported as a negative result.

#### 3.1.3. Sister Chromatid Exchange and Chromosomal Aberration

Double strand breaks (DSB) are the principal lesions in the development of chromosomal aberration (CA). SCE participates in the breakage of double-stranded DNA, followed by an exchange between homologous chromatids under the influence of any mutagen. Metaphase chromosome is the site to identify SCEs in the existence of 5-bromodeoxyuridine (BUDR) detected after two rounds of replication. The induction of SCE occurs during the S-phase of the cell cycle and is correlated with the recombinational repair of double-strand DNA breaks (DSB). Additionally, chromosomal aberration has been considered to be one of the important consequences of cells exposed to RF radiation systems. Changes in the chromosome structures and numbers are the signature of gene deregulation leading to genomic instability and cancer. Therefore, SCE and chromosomal aberrations may provide an indicator to study radiation-induced genotoxicity (Figure 2).

Maes et al. [100] have reported a marked increase in the frequency of chromosomal aberrations in human lymphocytes under the microwave exposure of 2450 MHz frequency for 30 min and 120 min with a SAR value of 75 W/kg, while no effect has been observed on the sister-chromatid exchange (SCE) at the same time [101].

Even with radiofrequency radiation exposure at 7700 MHz for 10, 30, 60 min in human blood lymphocytes, reported with significant elevation in the percentage of chromosomal aberrations (dicentric and ring chromosome) in the irradiated group (4.9%, 6.1%, 7.2%), as compared to non-exposed group (1.5%), confirming microwave radiation as a source for genomic changes in human somatic cells [102]. In vitro studies have been demonstrated with an increment in aberration frequency in human white blood cells after exposure at 954 MHz frequency to the blood sample for 2 h, SAR-1.5 W/Kg and even under RFR-exposure of 167 MHz frequency [100,103]. Microwave or ‘3G’ mobile telephony-radiation has also been reported to induce DNA damage and significant chromatid aberrations such as breaks (secondarily terminal deletions) and gaps (achromatic lesions) up to 275% in human cells, as compared to sham control [104,105]. Another study with 900 and 1800 MHz GSM—such as RF-EMF exposure— showed a significant direct genotoxic effect on human FCs (fetal cell) with increasing exposure time (3, 6 and 12 h), leading to cause delayed chromosomal condensation and significant rise in CAs [106].

However, controversial studies documented no significant changes in the amount of chromosomal damage after RF exposure at 2.45 GHz, 2.3 GHz, 1.8 GHz, 0.900 GHz, 0.820 GHz, 0.835 GHz, 0.847 GHz, 0.440 GHz, 0.380 GHz, 0.100 GHz in the human lymphocytes [107,108,109,110,111,112]. Zeni et al. [52] also reported no change in the frequency of sister chromatid exchange and chromosomal aberrations under the RFR GSM exposure at 900 MHz for 2 h, SAR 0.3 & 1.0 W/Kg in human peripheral blood leucocytes.

### 3.2. In Vivo Studies

In vivo studies provide data related to the interaction of radiofrequency radiation with biological systems, presenting a whole repertoire of body functions which is challenging to achieve with cellular studies. Differences in body size are considered an additional factor, demonstrating differences in dosimetric interaction according to the variable sizes. In comparison to humans, small animals represent the higher frequency and substantial penetration depth with respect to body sizes and their resonance to RFR. Most animal studies have been reported with somatic studies such as blood, bone marrow, brain, liver, or spleen. Only a few are dedicated to germ cells or the reproductive system to understand the mechanism of RFR. So far, many in vitro studies investigated with no direct genotoxic effect after acute or chronic exposure to RF-radiation [75,76,77,99,100,101,102,103,104,105,106,107,108,109,110,111,112,113]. To explore further, in vivo findings related to RFR studies on animal models are concluded under the following section of this review:

#### 3.2.1. DNA Damage

Apart from TUNEL, the comet is the most frequent and simple technique used to study DNA single and double-strand breaks after radiation exposure [114,115]. Comet assay is usually analyzed by tail moment, tail length, and tail intensity. With the help of this assay, authors reported a significant increase in the tail DNA percentage (138.03 ± 57.84 µm) and tail DNA moment (34.59 ± 45.02%) in the exposed group as compared to the sham exposed (39.96 ± 36.51 µm and 2.75 ± 3.08%), respectively, after whole body exposure of male Wistar rats to 3G [42]. The tail DNA percentage and tail DNA moment were also investigated to be increased significantly in the irradiated group as compared to control, after 2.4 GHz exposure [116]. Kumar et al. [117] further demonstrated a significant expansion in the tail intensity (15.1 ± 13.1%), tail length (154.4 ± 49.4 µm), and its moment (21.6 ± 14.7%) in sperm DNA after 10 GHz of microwaves exposure, as compared to control, where tail intensity (1.5 ± 2.01%), tail length (56.6 ± 14.2 µm) and tail moment (4.0 ± 0.5%) have been seen.

Aitken et al. [25] have reported significant DNA effects concluding genotoxicity in nuclear b-globin and mitochondrial genomes in caudal epididymal spermatozoa using RF exposure at 900 MHz and 1.7 GHz frequency in mice [118,119]. Houston et al. [120] demonstrated that after exposure of male mice to RF-EMF at 905 MHz frequency with SAR 2.2 W/Kg, 12 h/day for 1–5 weeks responded with 18% sperm DNA fragmentation during 1st week and significantly elevated after five weeks of exposure as compared to control or sham exposure populations. The damage to the DNA has been observed with single-strand breakage following whole-body radiofrequency exposure. The reporter suggested that the sensitivity of different germ cell populations after the in vivo RF-EMF experiment confounded the destructibility window to testicular and post-testicular phases of development. Apart from germ cells, many other tissues from rats and mice have also been reported with DNA damage after RFR exposure at 1900 MHz for 18 h/day, indicating the capability of RFR to induce genotoxicity [121].

Some authors have reported an indirect effect of DNA damage due to ROS generation after exposure to 900 MHz mobile phone radiation in Swiss albino mice [122]. A previous study from Pandey et al. [123] also supported a significant increase in DNA fragmentation with a frequency of 902.4 MHz, SAR-0.0516 W/Kg for 4 or 8 h/day. Additionally, exposure to 900 MHz EMF with SAR value 0.66 ± 0.01 W/Kg for 2 h/day for 50 days investigated with an increase in ROS generation that could trigger DNA damage due to activation of apoptotic genes and proteins (Bax, Bcl-2, cytochrome c, and caspase-3) involved in the signaling pathway after mitochondrial damage in rats [124]. One more report with exposure of male Wistar rats to 900 MHz RF-EMF (SAR-1.075 W/kg) found an alteration in MDA and ROS levels along with significant increase in DNA damage in the testicular tissue by 6.6 fold in tailed %, 2.2 fold increase in tail length and tail DNA, and 5.4 fold increase in a tail moment in comparison to control after comet assay examination [125]. A recent study by Mahmoud et al. [126] also demonstrated the harmful effects of cell phone exposure on spermatogenesis after exposure at 890–915 MHz (SAR 0.69 W/kg). A contradictory study has also been reported concerning short-term exposure. Guo et al. [127] demonstrated a marked increase in the levels of apoptotic proteins (Caspase 3, Bax) in testicular cells and disruption in the leydig cell function after 220 MHz pulsed modulated RF exposure. However, Dasdag et al. [128] reported no statistically significant alterations in testicular function or its structure after radiofrequency exposure at 250 MHz. Even after exposure to 1.5 GHz for 30 min at SAR 3, 6 and 12 W/kg, or short-term exposure at 900 or 1800 MHz at SAR 1.6 W/kg, reported with no significant damage to the reproductive system of a male mouse or rat [129,130].

Apart from short-term exposure, long-term exposure to cell phones could lead to degenerative alterations in testis [131]. Long-term exposure to 1800 MHz mobile phone radiation could lead to oxidative stress, which could directly promote the expression of BAX and stimulation of the p53 pathway, resulting in activating caspase 3 and hence testicular apoptosis [132]. Longer duration with higher RFR frequencies (1800 and 2100 MHz) resulted in a significant increase in DNA strand break in testicles [133]. Exposure to 4G smartphone suppresses male reproductive potential by disrupting Spock-3 testicular gene expression (Figure 2) [134]. Since electromagnetic wave energy is directly proportional to wave frequency, higher frequency results in more damage to the body tissue [135]. Based on the comet assay determination method, exposure at 2400 MHz frequency with SAR (0.11 W/Kg) for 24 h/day for 12 months concluded with a significant increase in the rat testes tissue in the experimental group as compared to sham control [116]. Meena et al. [41] communicated with a significant increase in the sperm DNA damage after whole-body exposure to microwave at the frequency of 2.45 GHz after measuring their tail length and tail moment using the comet assay. Kesari and Behari [61] also reported increased DNA fragmentation with cellular apoptosis for the same frequency (2450 MHz) after microwave exposure at a SAR value of 0.11 W/Kg (Figure 2).

All such studies may result in the accumulation of mutations that could lead to cancer formation in the next or subsequent generations.

#### 3.2.2. Micronuclei and Genomic Instability

Duration of exposure is a key factor in finding the intensity of DNA damage. Longer duration would result in more damaging effects, as compared to short-term exposure. Genomic instability could never result from short-term exposure [21]. Radiation-induced damage in the genome is denoted by an increase in the levels of genetic alterations in the progeny of irradiated group multiple generations after initial defamation [94,136]. Authors have reported an increase in the formation of micronuclei and genome instability after exposure to microwaves radiation [136]. Kesari et al. [96] found a significant increase in the ratio of PCE/NCE in the exposed group (0.67 ± 0.15), as compared with the non-exposed group (1.36 ± 0.07) after 35 days of mobile phone exposure in the rat sperm cells (Figure 2).

Micronuclei are used to determine chromosomal damage in rat’s bone marrow and peripheral blood erythrocytes after exposure to radiation [117]. The formation of micronuclei in bone marrow was reported to have a significant elevation after exposure to mobile phone at 0.9 W/Kg for 35 days [92]. Kesari et al. [137] also demonstrated a significant increase in the frequency of micronucleated polychromatic erythrocytes (PCE) in the irradiated group (132.66 ± 8.62 PCE/1000 erythrocytes) with respect to the sham-exposed group (15 ± 3.56 PCE/1000 erythrocytes). However, PCE/NCE (normochromatic erythrocyte) ratio by flow cytometry in blood cells was found to be significantly low after exposure to 3G mobile phone (0.67 ± 0.15), as compared to sham exposed (1.36 ± 0.007). Kumar et al. [117] on the other hand, communicated a statistically significant (*p* < 0.0004) increase by 52.75% in micronuclei formation in a blood sample after microwaves exposure (10 GHz) as compared to sham exposed (1.4 ± 0.4).

Micronuclei formation is directly proportional to the intensity of the damage. The chromosome fragments lost during cell division cannot be reversed or segregated to their opposite poles during the metaphase stage, causing genomic instability.

#### 3.2.3. Sister Chromatid Exchange and Chromosomal Aberrations

The pattern of responses in vivo reveals both positive as well as negative results at a frequency of 2450 MHz, with respect to chromosome translocations and sister chromatid exchange. Authors have been reported an increase in sperm cells abnormalities and SCE after 2 weeks of exposure at 2.45 GHz, in male CBA/CEY mice (Figure 2). However, controversial reports regarding sperm cells of male mice did not increase chromosomal aberrations at the same frequency (Table 2) [138,139].

Additionally, animals exposed to 100 W/m^2^ of 2.45 GHz continuous-microwave radiation for 6 h/day over 8-weeks concluded with no significant evidence of any chromosomal or SCE damage between sham and treated groups (exposed as stem cell spermatogonia) [140].

## 4. Genotoxicity and Oxidative Stress

Oxidative stress (OS) has been implicated as a significant source of infertility in men. It is a state that creates an imbalance between the levels of oxidants and antioxidants, causing the destruction of the biological system. If the rate of formation of free radicals will not be equal to their removal in the organism, then this will result in an impairment of the oxidative equilibrium, leading to oxidative stress, lipid peroxidation, and ROS formation. Antioxidants are known to neutralize such effects and thus help in mitigating infertility risks [141].

As compared to fertile control, infertile males showed a significant increase in seminal ROS levels with a decrease in antioxidant capacity [142,143,144,145,146,147]. RF-EMF underwent enhanced free radical generation in the exposed group, which could alter sperm and oxidative parameters like decrease in the levels of superoxide dismutase (SOD), catalase (CAT), total antioxidant capacity (TAC) or glutathione peroxidase (GSH-Px), and an increase in malondialdehyde (MDA) levels, affecting male reproduction against the ROS insult (Figure 2) [96,148,149,150,151,152]. ROS is considered an essential destructive agent in the production of genotoxic stress due to RF exposure. Schuermann & Mevissen [153] also investigated several experimental studies on animals and cells, showing elevated oxidative stress after RF-EMF exposure. Whole-body exposure to male Wistar rats at 2.45 GHz (SAR: 0.140 W/kg) for 2 h/day for 3 days demonstrated damage in spermatogenic cells and necrosis in seminiferous tubules under the induction of oxidative stress due to ROS [154]. Excessive ROS generation induces damage to DNA, RNA, and protein function in the spermatozoa along with other testicular cells [155]. Testicular OS has harmful consequences for male reproductive function. It brings about a reduction in the production of leydig cells or to the anterior pituitary [156,157]. Qin et al. [158] also reported damaging evidence to mouse leydig cells under the influence of OS after exposure to 1.8 GHz RF for 1, 2 and 4 h, which resulted in further reduction of testosterone production due to downregulation of clock genes (Rora, Clock, Baml 1) and its target gene expression (Star, Cyp11a1 and Hsd-3β) involved in testosterone synthesis (Figure 2).

Although DNA is considered a stable molecule, its interaction with free radicals eventually causes oxidative stress through various interaction mechanisms. ROS are a group of short-lived, highly reactive oxygen species that are well recognized to induce DNA damage by forming base adducts in DNA (forming 8-oxy guanine) [159]. RF-EMF also stimulates mitochondrial DNA lesions, DNA strand fragmentation, and mitochondrial DNA degradation under the influence of ROS-generated genotoxic stress. Microwave exposure was also reported to cause a significant increase in the formation of reactive oxygen species in sperm mediated by NADH oxidase in the plasma membrane, affecting male fertility (Figure 2) [14,94].

A recent study by Houston et al. [120] elucidated a significant increase in the mitochondrial ROS generation after 1-week exposure with an increased SAR value (905 MHz, 2.2 W/kg), causing elevated DNA oxidation and fragmentation. The author also reported an enhanced human spermatozoa ROS generation in mitochondria, resulting in the formation of DNA base adduct under the radiofrequency electromagnetic exposure [143].

Mobile phone ROS generation plays an essential role in causing genomic instability by inducing apoptosis, altering gene expression (such as Bax, cytochrome c, caspase 3), impairment in key protein functions due to protein folding, and production of stress protein (p38 MAP kinase) that phosphorylates heat shock protein (e.g., hsp 27) involved in sperm motility, and significantly reducing testosterone levels (*p* < 0.05) (Figure 2) [92,137,160,161,162].

Hou et al. [163] examined the effects of RF exposure at a frequency of 1800 MHz on mouse embryonic fibroblasts to study ROS, DNA damage, and apoptosis. The author has been investigated with an increase in the levels of both intracellular ROS and numbers of late-apoptotic cells in the RF-exposed groups for 1, 4 and 8 h as compared to control; however, the number of DSB has been found with a slight but no significant increase after 2, 4, 6 and 8 h of exposure in comparison to the untreated control group.

RF-EMF has been shown to disturb the intrinsic cellular antioxidant capacity by generating oxidative stress in many biological systems [164]. Furthermore, radiofrequency EMF exposure corresponds to DNA strand breaks that have been reported in spermatozoa and spermatocyte cells [30,73,91].

Such imbalance of ROS resulted in the reaction of hydroxyl radicals with DNA molecules due to the migration of hydrogen peroxide to the sperm head and targeting guanine residues in the 8th position within the sperm DNA, leading to cause base oxidation. 8-OHdG (8-hydroxy-2-deoxyguanosine) results from DNA base mutation and lesions, which could be a carrier for the next generation of the father’s germline, a consequence of the oxidative stress destruction of RF-EMF exposure (Figure 2).

The efficiency of the repair mechanism of DNA also admitted to being affected under the influence of ROS generation, and the precision of replication, as well as transcription, reported to be uncontrollable, emerging with changes in the DNA base structure and nucleotide loss and inaccurate ‘cross-link’ issues between DNA and molecules of protein (Figure 2) [165].

EMF waves are found to induce alterations in the cellular compounds of a cell (such as cell chromosome and chromatin material) by intervening genetic structure of the cell and its developmental cycle [166,167].

Moreover, aldehydes, which carry more reactivity capacity than free radicals, react instantly with the DNA molecule, causing cellular toxicity—including DNA damage and mutation [168].

## 5. Conclusions

The present review reveals a better understanding of the genotoxic effects of radiofrequency radiation on male reproductive health emitted from mobile phones, laptops, microwaves, wireless networks, etc. The study focused on different endpoints such as DNA damage, micronuclei formation and genomic instability, SCE & chromosomal aberrations covering both in vitro and in vivo parameters. The available information following in vitro and in vivo exposure shows that all the yielded data has both positive and negative results. In this review, studies reported DNA fragmentation, apoptosis, and elevated protein expression in both human and animal spermatozoa, concluding a decrease in viability, mitochondrial genomic destruction and DNA strand breaks. Further micronuclei formation, SCE and chromosomal aberrations are also found to cause abnormalities, leading to the accumulation of mutations and hence causing cancer risk. While controversial investigation, on the other hand, supported with no effect on cellular apoptosis or DNA integrity. Our present study reviewed that RFR has insufficient energy production to generate genomic damage. Yet, such effects were probably found to be responsible for male infertility due to the indirect mechanism of oxidative stress via ROS generation in the exposed system. Few studies also suggested that the damage due to the cumulative effect of repeated exposure varies with physical parameters such as distance from the radiation source, short-term or long-term exposure duration, penetration depth, and frequency of exposure. Therefore, considering all data together, the present review supports the capability of radiofrequency radiation to induce genotoxicity underlying male infertility keeping some limitations in mind, since the report is a conclusion of narrative study and limited literature were found explaining the actual mechanism of micronuclei formation, sister chromatid exchange, chromosomal aberration and genomic instability. Hence, more studies are needed to elucidate the DNA damage mechanism with more robust study designs favoring potential genotoxic effects of RFR on male reproductive health.

## Figures and Tables

**Figure 1 cells-12-00594-f001:**
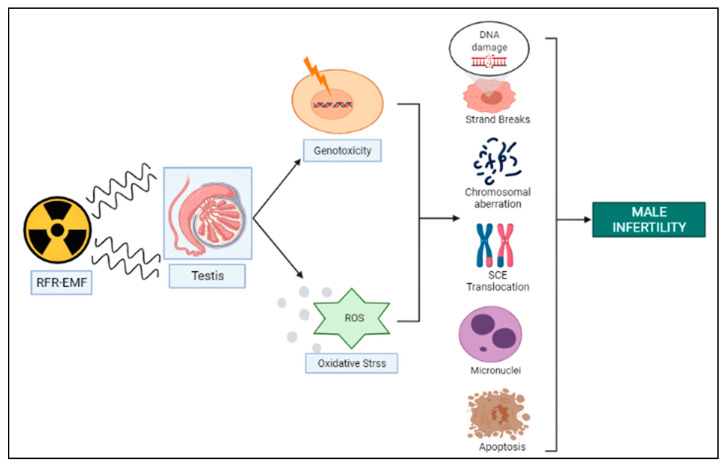
Possible effects of RF–EMF exposure on genotoxic parameters.

**Figure 2 cells-12-00594-f002:**
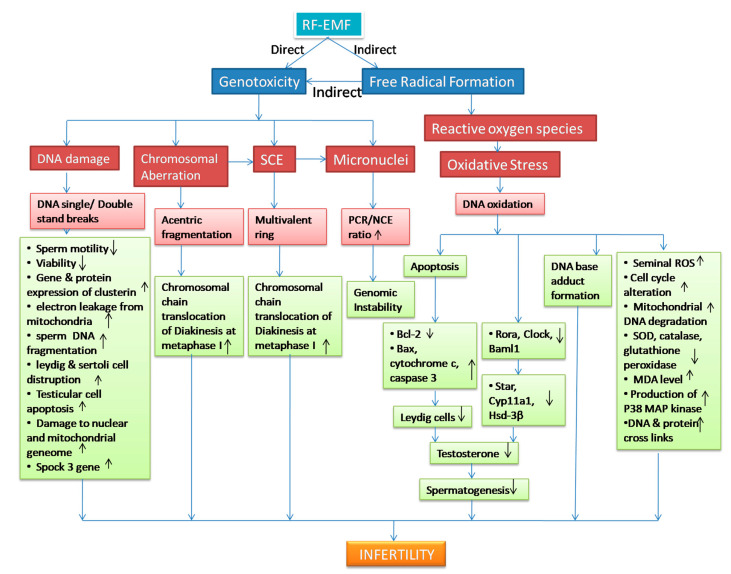
An overview of the genotoxic effects of RF-EMF on the male reproductive system. The figure suggests the probable radiation-induced damage, which may enhance ROS generation and genotoxic parameters such as DNA damage, micronuclei formation, chromosomal aberration, and SCE, leading to cause male infertility.

**Table 1 cells-12-00594-t001:** In vitro genotoxic studies on male reproductive system.

Genotoxic Endpoints	Subject	Frequency (MHz)	SAR	Dose Duration	Findings	References
DNA Damage	Human semen	850	1.46 W/kg	1 h	Significant increase in sperm DNA damage with a rise in gene and protein expression of clustering	[91]
DNA damage	Human semen	850	1.46 W/kg	1 h	No significant destruction in DNA integrity while an increase in ROS level reported	[77]
DNA Damage	Human semen	900	1.46 W/kg	1 h	Significant decrease in sperm motility and viability with the increase in DNA damage	[86]
DNA Damage	Human semen	900	2.0 and 5.7 W/kg	1 h	No significant induction of apoptosis in spermatozoa and no DNA fragmentation or any ROS generation	[71]
DNA Damage	Human semen	900/1800		5 h	Increase sperm DNA fragmentation with the decrease in sperm motility in exposed sperm	[85]
DNA Damage	Human semen	947.6	3.29 W/kg, 2.89 W/kg	3 h	Decreased SOD activity with a rise in DNA fragmentation and decline in sperm motility and viability with increase in oxidative stress	[88]
DNA Damage	Human Semen	947.6	3.29 and 2.89 W/kg	180 min	Significant increase in DNA fragmentation	[90]
DNA Damage	Human spermatozoa	1800	1.0 W/kg	16 h	Damage in DNA and sperm function due to electron leakage from the mitochondria and increased ROS generation, reduced motility and viability	[30]
DNA Damage	Cultured Mouse spermatocyte derieved GC-2-cell	1800	0.13 W/kg	1/20 min, 24 h	Accumulation of single stranded DNA break	[73]
DNA Damage	Mouse spermatocyte derieved GC-2-cell	1800	4 W/kg	24 h	Significant DNA damage via ROS generation	[74]
DNA Damage	Mouse spermatozoa	1800	0.15 W/kg & 1.5 W/kg	3 h	DNA fragmentation due to ROS generation under oxidative stress of RF exposure	[72]
DNA Damage	Human semen	Active mobile phone usage		More than 4 h/day	Sperm DNA fragmentation	[87]
DNA Damage	Mouse leydig cells	1950	3 W/kg	24 h	Cell proliferation inhibition, cell cycle alteration, dysfunction of testosterone secretion with no effect on ROS levels and cell apoptosis	[76]
DNA Damage	Human semen	2400		4 h	Sperm motility reduced progressively and sperm DNA damage increased. No significant difference observed in levels of dead sperm	[92]
DNA Damage	Human Semen	1800/2450		>30 min <121 min	Increased 8-OHdG expression and sperm nuclear DNA fragmentation. Sperm count, vitality, and motility decreased significantly with increase in oxidative stress	[93]

**Table 2 cells-12-00594-t002:** In vivo genotoxic studies on male reproductive system.

Genotoxic Endpoints	Subject	Frequency (MHz)	SAR	Dose Duration	Findings	References
**DNA Damage**	Male Sprague-Dawley rat	220	0.030 W/kg-whole body, 0.014 W/kg-testis	1 h/day, 30 days	Leydig and sertoli cell disruption along with cell apoptosis in testes	[127]
**DNA Damage**	Sprague-Dawley rat	250	0.52 W/kg	20 min/day, 1 month	No significant alteration in testicular functions (MDA concentration, sperm count, p53 immune reactivity)	[128]
**DNA Damage**	Male Wistar rat	890–915	0.69 W/kg	3 h/day, 2 weeks	Significant increase in apoptotic gene expression (caspase 3) and decrease in Bcl2, and significant decrease in sperm count, motility, viability, FSH, LH and testosterone with increase in MDA concentration	[126]
**DNA Damage**	Male Swiss mice	900	0.09 W/kg	12 h/day, 7 days	Significant damage to the mitochondrial and nuclear genome	[25]
**DNA Damage**	Male Swiss Albino mice	900	0.0054–0.0516 W/kg	6 h/day, 35 days	Increased DNA fragmentation and spermatogenesis arrest at the premeiotic stage due to increase in ROS generation	[122]
**DNA Damage**	Rat	900	0.66 ± 0.01 W/kg	2 h/day, 50 days	Significant increase in apoptosis due to elevated ROS levels and decreased TAC in sperm	[124]
**DNA Damage**	Male Wistar Rat	900	1.075 W/kg	2 h/day, 8 weeks	Elevated oxidative, inflammatory, apoptotic and testicular DNA damage	[125]
**DNA Damage**	Male Swiss Albino	902.4	0.0516 W/kg	4 or 8 h/day, 35 days	Significant increase in DNA damage	[123]
**DNA Damage**	Male C57BL/6 mice	905	2.2 W/kg	12 h/day, 1, 3 or 5 weeks	Elevated DNA oxidation and fragmentation (single strand break) and increased mitochondrial ROS generation after 1 week of exposure	[120]
**DNA Damage**	Male Swiss Albino	1800	0.05 W/kg	3 h/day, 120 days	Significant increase in testicular apoptosis due to elevated ROS levels with decrease in serum testosterone levels, sperm count and viability	[132]
**DNA Damage**	Male Sprague Dawley rat	1800/2100	0.166 W/kg, 0.174 W/kg	2 h/day, 6 months	Significant DNA single -strand fragmentation due to oxidative stress	[133]
**DNA Damage**	Male Wistar rat	1910.5	1.34 W/kg	2 h/day, 60 days	Increased MDA level and DNA strand break in sperm cells	[42]
**DNA Damage**	Male Wistar rat	2400	0.1 W/kg	24 h/day, 12 months	Significant increase in DNA damage in testes tissues	[116]
**DNA Damage**	Male Wistar rat	2450	0.14 W/kg	2 h/day, 45 days	Significant increase in sperm DNA damage, ROS, MDA, apoptosis, protein carbonyl content with decrease in testosterone level in testes	[41]
**DNA Damage**	Male Wistar rat	2450	0.11 W/kg	2 h/day, 35 days	Rise in DNA damage and cellular apoptosis	[61]
**DNA Damage**	Male Wistar rat	10,000	0.014 W/kg	2 h/day, 45 days	DNA strand break observed in sperm DNA in comet assay	[117]
**DNA Damage**	Rat testicular cells	4G	-	6 h/day, 150 days	Long term exposure impaired rat testis and unregulated testicular Spock-3 gene	[134]
**Micronuclei**	Male Wistar rat	900	0.9 W/kg	2 h/day, 35 days	Increase in micronuclei formation along with calatalse activity, MDA and ROS generation along with alteration in sperm cell cycle	[96]
**Chromosomal Aberration**	CBA/CEY male mice	2450	0.05–20 W/kg	30 min/day, 6 days/week, 2 weeks	Significant increase in sperm cell chromosomal chain translocation observed at diakinesis at metaphase I	[138]
**Chromosomal Aberration**	Male mice	2450	-	30 min/day, 6 days/week, 2 weeks	No increase in sperm cell chromosomal aberrations	[139]
**SCE**	CBA/CEY male mice	2450	0.05–20 W/kg	30 min/day, 6 days/week, 2 weeks	Significant increase in sperm cell chromosomal chain translocation observed at diakinesis at metaphase I	[138]

## Data Availability

Not applicable.

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
