# Peer review of "Genotoxic Risks to Male Reproductive Health from Radiofrequency Radiation"

_cells, 2023, doi:10.3390/cells12040594_

Round 1
Reviewer 1 Report
Comments on the manuscript “Genotoxic Risks to Male Reproductive Health from Radiofrequency Radiation”.
This is a well-prepared manuscript and due to the technological advancement in electronic gadgets, the study question should be addressed.
The abstract reads well.
The last part of the introduction is more like a review, which could be limited.
Section 3.1.1 the table is very information so the description part is repetition of the results and could be significantly reduced.
Similarly in section 3.2.1.
One drawback in this study is that the authors could have emphasized the mechanism causing the genotoxic effect.
Author Response
Authors are grateful to the reviewer's valuable comments and suggestions.
Point-by-point response to the reviewer’s comments has been attached.
Please see the attachment.

Reviewer 2 Report
In this manuscript, Kaur and Singh provided a comprehensive review of genotoxic risks to male reproductive health from radiofrequency radiation. The authors summarized both in vitro and in vivo studies, along with a focus on oxidative stress in this context and discussed the previously contradictory findings. The structure is well-organized and easy to follow. Overall, this review paper is nicely written. Once published, this review will provide the field with a good reference of what is known and what yet to be discovered. Before acceptance of this manuscript, I encourage the authors to carefully double-check the manuscript and eliminate all the grammatical errors and typos (some examples include L41, L208 and L408).
Minor suggestions:
1) In L98, better specify till which month of 2022 the articles are incorporated.
2) L133, either there’s no need to write the full name of ‘DNA’ here, or please move it to the first time the term ‘DNA’ was introduced in the manuscript.
Author Response
Authors are grateful to reviewer for their valuable comments and suggestions.
Point-by-point response to the reviewer’s comments has been attached.
Please see the attachment
